# What’s New in *Cryptococcus gattii*: From Bench to Bedside and Beyond

**DOI:** 10.3390/jof9010041

**Published:** 2022-12-27

**Authors:** Justin Beardsley, Aiken Dao, Caitlin Keighley, Katherine Garnham, Catriona Halliday, Sharon C.-A. Chen, Tania C. Sorrell

**Affiliations:** 1Sydney Infectious Disease Institute, University of Sydney, Sydney, NSW 2145, Australia; 2Westmead Hospital, New South Wales Health, Sydney, NSW 2145, Australia; 3Westmead Institute for Medical Research, Sydney, NSW 2145, Australia; 4Sunshine Coast University Hospital, Sunshine Coast University, Birtinya, QLD 4575, Australia; 5Institute of Clinical Pathology and Medical Research (ICPMR), NSW Health Pathology, Sydney, NSW 2145, Australia

**Keywords:** *Cryptococcus gattii*, cryptococcosis, fungal infection, medical mycology, diagnostic tools, epidemiology, antifungal, antimicrobial resistance

## Abstract

*Cryptococcus* species are a major cause of life-threatening infections in immunocompromised and immunocompetent hosts. While most disease is caused by *Cryptococcus neoformans*, *Cryptococcus gattii*, a genotypically and phenotypically distinct species, is responsible for 11–33% of global cases of cryptococcosis. Despite best treatment, *C. gattii* infections are associated with early mortality rates of 10–25%. The World Health Organization’s recently released Fungal Priority Pathogen List classified *C. gattii* as a medium-priority pathogen due to the lack of effective therapies and robust clinical and epidemiological data. This narrative review summarizes the latest research on the taxonomy, epidemiology, pathogenesis, laboratory testing, and management of *C. gattii* infections.

## 1. Introduction

*Cryptococcus* can cause life threatening infections in both immunocompromised and immunocompetent hosts. Although most disease is caused by *Cryptococcus neoformans*, *C. gattii*—a genotypically and phenotypically distinct species—causes 11–33% of invasive cryptococcosis globally [1,2,3]. Even with best available therapy, *C. gattii* is associated with 28 day case fatality rates in range 10–25% [4,5,6,7]. The World Health Organisation’s recently released Fungal Priority Pathogen List categorised *C. gattii* as a medium priority pathogen in light of the poor availability of safe and effective therapies and a lack of robust clinical and epidemiological data [8].

We conducted a narrative review of the literature, summarising what is new in epidemiology, pathogenesis, diagnosis, and management of *Cryptococcus gattii* infections—contrasting this, where relevant, with what is known about the more common pathogen, *Cryptococcus neoformans*. Data, as far as possible, are from the last 10 years. Only English-language publications were included.

## 2. Latest on Taxonomy and Molecular Epidemiology of Cryptococcus

*Cryptococcus neoformans* and *Cryptococcus gattii* are haploid encapsulated yeast-like fungi of the class *Trellomycetes* [9]. First described in 1895, there have been numerous revisions in nomenclature since [9]. *Cryptococcus neoformans* includes two varieties, *Cryptococcus neoformans* var. *grubii* and *Cryptococcus neoformans* var. *neoformans*, though further sub-divisions have been proposed; *Cryptococcus gattii* is a separate species [9,10,11]. The term species complex is used for both in recognition of the substantial genetic variation within the *Cryptococcus neoformans* and *Cryptococcus gattii* complexes [12].

Historically, *Cryptococcus* spp. were distinguished by serotype with serotypes A, B and C recognized two decades before serotype D [9]. Serotyping initially used rabbit sera to react with capsular polysaccharides before moving on to PCR targeting the laccase gene (*LAC1*) and capsule gene (*CAP64*) [13,14,15]. Serotypes A and D are both *Cryptococcus neoformans*: A is *C. neoformans* var *grubii* and D is var *neoformans*. Serotypes B and C are *Cryptococcus gattii*. Hybrids also exist. A minority of *Cryptococcus* spp. (ranging from 2–21%), are acapsular or pauci-capsular, and so untypeable [16,17,18].

The advent of molecular taxonomy has revolutionized fungal nomenclature. Various techniques have been used including PCR-fingerprinting, Restriction Fragment Length Polymorphism (RFLP), Amplified Fragment Length Polymorphism (AFLP) and Multi-Locus Sequence Typing (MLST) to identify distinct lineages within the two species complexes: *C. gattii* (VGI-VGIV) and *C. neoformans* (VNI, VNII, VNB) (now separated into VNBI and II–all serotype A), VNIII (hybrid of serotypes A and D), VNIV–serotype D) [19] (see Table 1).

Adding a further genotype based on a single clinical isolate from Mexico, Hagen et al. in 2015 proposed that *C. neoformans* and *C. gattii* be divided into 7 sibling species based on the phylogenetic (evolutionary) species concept. Namely, for *neoformans*: *Cryptococcus neoformans* (genotype VNI), *Cryptococcus deneoformans* (genotype VNIV), plus *C. neoformans* × *C. deoneoformans* hybrid (genotype VNIII). For *gattii*: *Cryptococcus gattii* (genotype VGI), *Cryptococcus deuterogattii* (VGII), *Cryptococcus bacillispora* (VGIII), *Cryptococcus tetragattii* (VGIV), and *Cryptococcus decagattii* [20]. Subsequently, a genetically distinct lineage of *C. gattii* discovered in Zambian woodland environments in association with nitrogen-rich middens of a small mammal known as Hyrax, has been designated VGV, and *C. decagattii* as VGVI [21,22].

Debate continues within the scientific community as to whether *C. neoformans* and *C. gattii* should be considered species complexes, supported by clinical, epidemiological and ecological data, or divided into separate clades (species) with corresponding changes in nomenclature, based on genotyping alone, without clearcut biological differences [12,20,23]. It was argued by Kwon-Chung and colleagues in 2017 [12] and later by Farrer et al. [21] that species designations as proposed are not yet stable. The recent characterization of multiple environmental isolates of a new *C. gattii* lineage in Zambian woodlands introduced above suggests that further lineages will be discovered [21].

MLST-based DNA Barcoding, which uses short diverse genetic sequences that are stable and unique to a given species, is currently the gold standard for cryptococcal typing [21,24,25]. A multitude of genetic loci were initially proposed for MLST, complicating its utility and comparability [9,22]. A single schema is now supported by the International Society for Human and Animal Mycology (ISHAM) and includes seven loci (*URA5*, *CAP59, GPD1, LAC1, SOD1, PLB1*, and *IGS*). A database of quality-controlled reference sequences is available at https://mlst.mycologylab.org/ (accessed on 30 June 2022). and in the CBS-KNAW database (Westerdijk Fungal Bioversity Institute, https://www.knaw.nl (accessed on 30 June 2022)). As of 30 June 2022, this data base contained 566 entries for *C. gattii* and 662 for *C. neoformans*.

Accurate species identification from fungal colonies can also be achieved reliably and rapidly by matrix-assisted laser desorption/ionization-time of flight mass spectrometry (MALDI-TOF MS) [25] with minimal sample preparation or it can be combined with nucleic acid-based testing. Using MALDI-TOF MS methods, *C. neoformans* and *C. gattii* can be distinguished readily [24,26,27]. However, because of lack of standardization of sample preparative procedures and limitations of commercial databases, the use of MALDI-TOF MS for detailed fungal identification is generally limited to reference or research laboratories.

**Table 1 jof-09-00041-t001:** Classification of *Cryptococcus neoformans* and *Cryptococcus gattii* species complexes based on serotype, molecular sub-type, and genomic sequence.

Species Complex	Proposed Species Name	Variety	Serotype	PCR Fingerprinting	Reference Strain—Accession ID of Reference Assembly
*Cryptococcus neoformans* species complex	*C. neoformans*	var. *grubii*	A	VNI	H99-ASM301198v1
				VNB	Bt88-BROAD_CneoA_Bt88_1
				VNII	PMHc1023.ENR BROAD_CneoA_PMHc1023.Enr_1
	*C. deneoformans*	var. *neoformans*	D	VNIV	JEC21-ASM9104v1
	*C. neoformans* × *C. deneoformans hybrid*	AD hybrid	AD	VNIII	
*Cryptocccus gattii* species complex	*C. gattii*		B/C	VGI	WM276-ASM18594v1 Ru294-Cryp_gatt_Ru294_V1
	*C. deuterogattii*		B/C	VGII, VGIIa, b, c, d	R265-R265.1
	*C. bacillisporus*		B/C	VGIII	CA1280-Cryp_gatt_CA1280_V1
			B		CA1873-Cryp_gatt_CA1873_V1
	*C. tetragattii*		B/C	VGIV	IND107-Cryp_gatt_IND107_V2
			B		MF34-Cryp_gatt_MF34
	-		B/C	VGV	
	*C. decagattii*		B/C	VGVI	

Adapted from Hong 2021 Frontiers in Cellular and Infection Microbiology [28].

MLST has revealed geographical associations with clonal populations with increased heterogeneity possibly related to the origins of the species; for example, *C. gattii*, may have spread from Australia and Northern Brazil to North America and Canada [28,29] whereas for *C. neoformans,* ST5 is the predominant serotype in China and Taiwan, ST93 in Brazil, ST41 in Japan and the US, with greater diversity in African countries [16,28,29,30,31,32].

Despite the value and comparatively low cost of MLST, the highest resolution is achieved with whole genome (next-generation) sequencing (WGS or NGS). This allows greater discrimination between strains and offers a method for species identification, detection of drug resistance mutations and epidemiological information. WGS has broadly confirmed the relationships between the different genotypes identified by MLST. The first whole genome sequence for *C. neoformans* was published in 2005 [33]. There are now 115 genomes incorporated in NCBI belonging to what was previously known as *C. neoformans* (NCBI accessed 30 June 2022). This has enabled further elucidation of origins, evolution, and genetic variation amongst the species complexes [13,34]. Associations between genomic differences and virulence or habitat have been postulated [35,36]. Because of the tendency of these yeasts towards aneuploidy and to become diploid, as well as more minor structural rearrangements, accuracy requires a higher genome coverage than that accepted for simpler organisms, and complete assessment may require long read sequencing to supplement short read sequencing [28,37]. Nonetheless, WGS enabled the delineation of VGV *C. gattii* isolates in Zambia, and sub-lineages VNBI and VNBII within VNB of sub-Saharan African isolates associated with phenotypic differences, as well as the new designation of VGVI (putative *C. decagattii*) [19,20,21].

In summary, debate persists regarding the delineation of species within both *C. gattii* and *C. neoformans* species complexes. For clinical purposes, the use of species complexes is the more useful terminology and will be used from this point. Meanwhile, important research continues into the implications of different genetic groupings.

## 3. Ecology, Epidemiology and Clinical Features

*C. gattii* and *C. neoformans* are adapted to different ecological niches and exhibit differences in epidemiology, clinical features, and virulence mechanisms [38,39].

### 3.1. Ecology

Several arboreal species have been identified as the predominant environmental reservoir of *C. gattii*, often eucalypts but including a huge range from oaks to baobabs [40]. *C. neoformans*, on the other hand, is primarily associated with weathered pigeon guano.

Despite potential sampling bias across the globe, evidence to date indicates that *C. gattii* infections occur predominantly in Australia, British Columbia, Canada, the Pacific Northwest of the USA, parts of South America and Africa. Molecular type VGI, has a global distribution but is most prevalent in Africa, Australia, and Europe. Molecular type VGII occurs globally but is mostly reported from Australia and the Americas. Molecular type VGIII is concentrated in, but not limited to, the Americas. The relatively rare molecular type VGIV has been reported in southern Africa and India, while the very rare molecular type VGVI has been reported from Mexico [9,20,21,40,41].

There is some evidence that subgroups (subclades) within the VG lineages are also geographically defined, for example VGIa is found in Australia and the Pacific Northwest of the US, whereas VGI b has been found in Zambia; VGII a and c predominate in the Vancouver area and the Pacific Northwest of the US whereas VGIIb has been reported in far Northern Australia. These associations may reflect microevolution within these lineages over time.

Interest has recently turned to environmental factors affecting the distribution and virulence of fungal species, including *C. gattii.* Modelling performed by Cogliati et al. determined that in Europe, *C. gattii* was concentrated along the Mediterranean coast, with its distribution limited by low temperatures during the coldest season, and by heavy precipitations in the driest season; *C. neoformans* var. *grubii* colonized the same areas but it tolerated cold winter temperatures and summer precipitations better [42]. More extensive modelling using Maxent analysis showed a gradual expansion of *C gattii* from 1980 to 2009 in the same region followed by doubling of the potential environmental niche in 2010–2019 and a predicted further extension inland from the coastal Mediterranean basin as a result of climatic change and global warming [43].

In relation to evolution of *C. gattii,* the discovery of VGV coexisting with VGI and VGII in Zambia in an entirely new ecological niche, namely, middens of the Southern tree hyrax (*Dendrohyrax arboreus*) [21] is of great interest. Hyrax middens concentrate faecal material and are extremely stable and long-lasting structures that can exist in the same place for thousands of years [44]. They have a high nitrogen content which is known to aid cryptococcal growth. Hence due to their extreme environmental stability, these middens may provide stable, long-term, evolutionary niches that facilitate the evolution of genetic diversity within *Cryptococcus*.

### 3.2. Epidemiology

Although it was traditionally believed that *C. gattii* infection occurred predominantly in immunocompetent people, evidence increasingly points to potential immune and genetic factors predisposing to infection. In Australia the incidence of cryptococcosis in aboriginal populations is 10.4/million/year compared with 0.7/million/year in non-indigenous populations, and the difference cannot be wholly explained by differences in places of domicile or work [45]. In the outbreak centred on the Pacific Northwest of the USA, some form of immune deficiency or underlying health problem has been identified in approximately half of patients [46,47]. Subtle immune defects can play an important role. For example, clear links have been established between high titres of autoantibodies to granulocyte-macrophage colony-stimulating factor and both acquisition and severity of *C. gattii* infection [48,49]. Over the last five years, a plethora of new immunological and genetic factors predisposing to cryptococcosis have been described—more research is needed to understand their role in *C. gattii*.

When dealing with *C. gattii* infections, clinicians must consider the possibility of an underlying immune defect. Although many major causes of immunocompromise may be obvious on history taking, such as transplant or haematological malignancy, others require specific questioning about history of prior infections, family history of unusual infections, early deaths and/or malignancies. Positive findings may indicate further testing is desirable, such as blood smear/film, CD4/CD8/natural killer (NK) cell subsets by flow cytometry, immunoglobulin levels, complement testing and others. Where inborn errors of immunity are suspected, genetic testing will be required, and a clinical immunologist should be involved in discussions.

### 3.3. Clinical Features and Diagnosis

Although *C. gattii* is less likely to present with CNS involvement (varying by genotype), when it does, there is a higher incidence of CNS imaging abnormalities, mass lesions and hydrocephalus requiring neurosurgical intervention. There appear to be significant differences in cerebrospinal fluid (CSF) examination between the two species, with *C. gattii* CNS infections demonstrating lower CSF glucose and protein, higher median white blood cell counts and more frequent India ink positivity. Immune reconstitution inflammatory syndrome (IRIS) is more frequent in *C. gattii* infections than *C. neoformans*. Despite all this, CNS *C. gattii* infection has a lower mortality rate than *C. neoformans*. It must be noted, however, that mortality in patients with cryptococcal infections is often related to the predisposing disease rather than direct cryptococcal-induced mortality and this may partially explain this observation [6,39]. See Table 2 for a summary.

## 4. Advances in Understanding of Virulence and Pathogenesis

After inhaled Cryptococcus cells are phagocytosed by alveolar macrophages, they alter the macrophage transcriptome, preventing significant acidification, calcium efflux and protease activity, and thereby allowing intracellular pathogen proliferation [51].

Classically activated M1 macrophages that result from the usual Th1 cytokine immune profiles are essential for clearing Cryptococcus, while alternatively activated M2 macrophages resulting from Th2 cytokine stimulation are associated with a higher burden of disease [52,53]. *C. gattii* can modulate macrophage polarisation to M2, thereby evading immune recognition and clearance [51,52,53]. There are observed species-dependent in vitro differences in M1-polarised macrophage transcriptomic gene regulation and subsequent bioprocess modulation. Both Cryptococcus species affect the Akt/mTOR pathway and TNF-alpha gene expression, reducing protective M1 macrophage fungicidal activity. Subsequently, there is dysregulation of normal immunologic dendritic cell development, NK cell activation and proliferation, other cellular cytokine profiles and nitric oxide production, leading to the alternative Th2-activated M2 macrophage polarisation [51]. *C. gattii* impacts signalling, cell differentiation and regulation of immune system processes, while *C. neoformans* modulates the cellular response to stimulus, response to DNA damage and cell death [51].

Deficiencies in alveolar macrophage functions, including phagocytosis and pattern recognition receptors, result in an intracellular proliferation of Cryptococcus. Subsequent abnormalities in cytokine and protein signalling pathways then initiate a suboptimal or potentially pathogenic immune response, depending on where the deficiency lies [53]. Advances in defining these pathways have led to diagnoses of immunodeficiencies in otherwise seemingly healthy hosts and stimulated significant interest in therapeutic approaches.

Endogenous interferon (IFN)-gamma production is associated with protection against Cryptococcus disease, leading to increased phagocytosis and fungicidal activity of phagocytes. Although exogneous IFN-gamma is sometimes useful as a therapy for refractory IRIS [54], as discussed in the treatment section, it has not been proven effective as a treatment adjunct for cryptococcal meningitis. Anti-granulocyte-macrophage colony-stimulating factor autoantibodies, introduced as a predisposing factor above, result in defective surfactant clearance by pulmonary macrophages [55] offering a mechanistic explanation for increased risk of infection, although as yet this has not been translated into a therapuetic option.

*C. gattii* VGI and VGII genotypes cause most infections in immunocompetent hosts, while VGIII and VGIV are mainly isolated from immunocompromised hosts. Genotype VGII is the most heat-resistant of the known strains, especially at 37 °C. Different subtypes have different gene expressions identified in vitro, which correlate with their observed virulence clinically. Notably, the outbreak strain VGIIa, in a whole genome sequencing study, demonstrated genetic transformation and mitotic microevolution from 12 missense mutations and one shift mutation. Hypervirulent strains of *C. gattii*, but not *C. neoformans*, demonstrate enhanced mitochondrial tubularisation, which positively correlates with intracellular proliferative capacity within macrophages, a possible species-specific virulence mechanism [56].

Both species elicit a robust host immune response. However, murine studies note a considerably different species-dependent host transcriptomic profile [52]. There is variation in the molecular and chemical properties of extracellular enzymes secreted by *C. gattii* compared with *C. neoformans*, perhaps reflecting their different environmental niches and contributing to different clinical manifestations [57].

Another difference is related to the cryptococcal capsule. The cryptococcal capsular polysaccharides, glucuronoxylomannan (GXM) and galactoxylomannan, simultaneously evoke an immune response, enhancing pathogen virulence and allowing evasion of immune recognition by coating antigens on *C. gattii* [58]. The GXM of *C. gattii* has an additional xylose residue compared to *C. neoformans* [59].

Two structural factors associated with increased virulence, the larger size of the dynamic capsule and increased melanin production protecting against reactive oxygen species, are impacted by environmental conditions, with the former shown to be altered by some common herbicides [60]. The different environmental niches of *C. neoformans* and *C. gattii*, hypothetically, are subjected to different herbicidal and environmental pressures, which may differentially alter capsule size and melanin production.

Capsule size correlates with virulence, although not case-fatalit [61]. Notably, *C. gattii* seems to have a greater propensity to form titan cells [62]. Titan cells are abnormally sized (>10 µm) yeast cells that have an immunomodulatory role and potential roles in the pathogenesis of cryptococcosis [63].

*C. gattii* melanise more slowly than *C. neoformans* [61]. Melanin modulates susceptibility to amphotericin and fluconazole [61], and the speed of melanisation may correlate with virulence and subsequent morbidity and mortality, making this another important avenue of investigation into differences in disease phenotype between the species. It is important to note that there is no evidence showing a relationship between susceptibility and treatment outcome.

Some strains of *C. gattii* have a greater ability to form biofilms in vitro, which may also contribute to differences in disease phenotype between the species [64]. Along with higher levels of the enzymes phospholipase and haemolysin activity, the ability to form biofilms has been associated with human pathogenicity, and correlates with histopathologic pulmonary tissue damage in animal models [64,65,66].

A recent in-depth review summarising *C. gattii* genotypes, phenotypes, virulence and regulatory mechanisms is recommended for those interested in further reading on this topic [57].

## 5. Novel Therapeutics for *C. gattii*

Six international consensus guidelines have been released in the past decade covering cryptococcal infections [67,68,69,70,71,72]. None are targeted explicitly at *C. gattii* infection, and recommendations are based on expert opinion as, to date, no prospective randomized controlled treatment trials have been conducted for *C. gattii* infection. The rarity of *C. gattii* infection presents a significant challenge to conducting such trials. For severe infection, many clinicians favour a prolonged induction therapy with liposomal amphotericin B and 5-flucytosine (4–6 weeks) followed by consolidation therapy for 12–18 months, and this is supported by an Australian case series [5]. There have been no significant new data to guide best use of existing antifungal therapies in *C. gattii* infections in the last decade—more research is needed to optimise this aspect of care.

A comprehensive review of the antifungal drug pipeline in 2021 [73] reported the spectra of activity for fosmanogepix, ibrexafungerp, olorofim, opelconazole, and rezafungin (all agents in late-stage clinical development). Fosmanogepix and opelconazole have potent in vitro activity against *C. gattii* and *C. neoformans*. Olorofim has no activity against either. Rezafungin, as consistent with data for the echinocandin class of antifungal, is not active against *C. neoformans* and is unlikely to be effective for *C. gattii* (although data are awaited). There were no data for ibrexafungerp. Not included in that review was oteseconazole, a novel tetrazole, which shows activity against *C. neoformans*, though no data are available for *C. gattii* [74,75]. A less developed agent, T-2307, also shows evidence of activity against both species in vitro and in animal models [76,77]. Several other agents are undergoing pre-clinical investigation for activity against *C. neoformans* but not *C. gattii*.

Given the relative dearth of agents in the antifungal pipeline and the ongoing high mortality and morbidity rates with existing antifungal therapy, researchers have enthusiastically pursued adjunctive therapies. Despite their theoretical promise, trials have returned disappointing results. However, it must be highlighted that the following studies for “cryptococcal meningitis” were not species-specific and mainly recruited *C. neoformans* patients.

Both pathogen and host play roles in tissue damage, so immunomodulation is a natural target for adjunctive therapies. Under this category, both IFN-gamma and dexamethasone have been subject to clinical trials in HIV co-infected patients. In 2004, a double-blind placebo-controlled trial (n = 70) of adjunctive IFN-gamma Ib (100 or 200 μg thrice weekly for ten weeks) showed a non-significant trend towards improved CSF clearance of cryptococci [78]. However, adverse events increased significantly. In 2012, a randomised, open-label study in Malawi (n = 88) (again in HIV co-infection) compared two/six adjunctive doses of 100 μg IFN-gamma Ib. Although patients receiving IFN-gamma cleared cryptococci from their CSF more quickly, there was no mortality benefit [79]. There are currently no further trials registered for IFN-gamma at ClinicalTrials.gov as of 26 November 2022.

Dexamethasone was considered a promising therapy for HIV-associated CM based on mortality benefits in tuberculous meningitis and confirmed bacterial meningitis [80], and supportive evidence from cryptococcal meningitis observational studies (particularly in *C. gattii*) and animal models. In 2016, results of a double-blind, randomised placebo-controlled trial (“Crypto-Dex”, n = 451) which recruited patients in Vietnam, Thailand, Indonesia, Uganda, and Malawi, were released [81]. The trial was stopped early by the data safety and monitoring board as clear evidence emerged that the same dose of dexamethasone was used successfully in tuberculous meningitis (starting at 0.3 mg/kg/day and weaning weekly over six weeks to 1 mg/day) was harmful in this group [81]. Dexamethasone caused higher rates of disability, slower fungal clearance, increased frequency of adverse events, and a statistically non-significant increase in 10-week mortality (47% vs. 41%) [81]. Notably, the hazards were non-proportional, with early signs of benefit outweighing later signs of harm. It remains unclear whether steroids could be helpful in specific subgroups (e.g., space-occupying lesions), whether a shorter course at induction would have led to a different outcome, or indeed whether outcomes would differ for patients with *C. gattii* infection.

Unfortunately, there is insufficient evidence to inform clinicians about the harms or benefits of steroids in HIV-uninfected cryptococcal meningitis patients. These patients have a very different baseline immune response to HIV-co-infected patients, so responses are likely to differ. Case series have described benefits, especially in *C. gattii* infections and patients with mass CNS lesions [82,83], but there is no robust trial evidence. Recent guidelines recommend against general use but support short courses in specific indications such as space-occupying lesions with surrounding mass effect [84].

Other agents have been identified as promising non-immunomodulating adjuncts, although again, results have been disappointing. Between 2018–2020 three trials were published reporting on the efficacy of the selective serotonin reuptake inhibitor sertraline in cryptococcosis [85,86,87]. A significant phase III double-blind, randomised placebo-controlled trial (n = 460) in Uganda [85] and in a smaller study in Mexico (n = 12) [86] both targeted HIV-associated CM and showed neither microbiological nor clinical benefit. Another trial looked at the efficacy of sertraline in HIV-associated asymptomatic antigenaemia but was ceased after just 21 patients due to an excess of adverse events [87]. Based on in vitro evidence for efficacy, adjunctive tamoxifen 300 mg/day was trialled in a randomised open-label trial (n = 50), but results published in 2021 show no benefit [88].

Work on cryptococcal vaccinations has been undertaken since the 1950s, with many different antigens and delivery systems protecting against infection or attenuation of disease in animal studies. More recently, acapsular strains of *C. gattii* and dendritic cell-based vaccines are under investigation as vaccine antigens [58,89,90]. Adjuvants are likely to play an important role in stimulating an immune response that is effective at multiple sites without causing undue toxicities. They are the subject of intense investigation, as summarised in a 2021 paper by Oliveira et al. [91]. Although vaccines for some fungal pathogens have reach clinical trials, none directed against Cryptococcus have yet reached this stage. Recently, chimeric antigen receptor (CAR) cytotoxic T cells targeting glucuronoxylomannan have been demonstrated in vitro to bind to *C. gattii* and *C. neoformans* regardless of whether the cells are titan or not, but with a more pronounced affinity for the yeast form and reduced the size and number of pulmonary titan cells in cryptococcus infected mice [63,92].

## 6. Updates in Diagnostics 

Although there are multiple strategies to diagnose cryptococcosis, with and without fungal culture, diagnosis can be challenging. Delays in treatment because of missed early diagnoses or incorrect diagnoses, lead to worse clinical outcomes [93].

Traditionally, fungal culture has been used to diagnose cryptococcosis (see Figure 1). Whilst reasonably sensitive, this requires an equipped laboratory and is time consuming [94]. The species complexes can be distinguished by their growth on canavanine-glycine-bromthymol blue (CGB) agar; *C. gattii* turns the culture medium blue, whereas *C. neoformans* does not change the colour of the medium [95].

In terms of microscopy, Cryptococcus can be seen in body fluid with India ink (see Figure 1) examination, histopathology of infected tissue with specific stains to identify capsule (mucicarmine and alcian blue) or presence of Fontana Masson’s melanin detection [93] in addition to standard histopathological stains. Although the India ink detection method of cerebrospinal fluid (CSF) is simple and fast, the sensitivity is only 50% and it is unable to distinguish between the two species complexes.

X-ray, computerised tomography (CT) scans and magnetic resonance images (MRI) of the lungs (pulmonary cryptococcosis), brain (cryptococcal meningoencephalitis), and other parts of the body are valuable adjunctive tools for diagnosing cryptococcosis and especially the extent of disease, but radiography results alone are unlikely to be diagnostic [96,97,98].

Serological diagnostic tests are used to detect cryptococcal capsular polysaccharide in serum and CSF by Latex agglutination test (LATs), enzyme-linked immunoassays (ELISA). These tests have an overall sensitivity and specificity of 93–100% and 93–98%, respectively [93]. The false positive rate is less than 1% mostly due to technical issues or other infections (e.g., a cross reaction with antigens from Trichosporon species). A possible limitation is that LATs based on monoclonal antibodies can be less sensitive to *C. gattii* infections, including infections caused by *C. gattii* and *C. neoformans* hybrids [99].

The introduction of cryptococcal antigen lateral flow assays (CrAg-LFA) marked a major revolution in the diagnosis of cryptococcal infections [100]. CrAG-LFA is a dipstick immunochromatographic assay that is simple to use and does not require any pathology laboratory infrastructure (see Figure 2) [101]. Additionally, the LFA shows excellent concordance with Latex Agglutination, ELISA and cultures. It works well in resource-limited settings and effectively diagnoses *C. gattii* infections which may be missed by other serological tests.

Although molecular assays, such as polymerase chain reactions (PCRs), are potentially a more rapid diagnostic method than culture and antigen testing, such assays are not routienly required. When in use, commercial PCR-based assays that target Cryptococcus cannot distinguish between *C. neoformans* and *C. gattii*. Recently, a real-time PCR assay targeting the multicopy mitochondrial cytochrome b (cyt b) gene to detect *C. neoformans* and *C. gattii* has been developed [102]. The assay was tested in clinical specimens, showing that the cyt b-directed assay accurately detected and identified all eight molecular genotypes of *C. neoformans* and *C. gattii*. The overall reported assay sensitivity was 96.4%, and the specificity was 100%, and it can diagnose cryptococcosis in patients within four hours [102]. The targeted assay is cost-saving (USD 40 per sample) and applicable to a diverse range of clinical specimens, including respiratory tract specimens and formalin-fixed paraffin-embedded tissue that is not feasible for culture.

The novel lateral flow strips combined with recombinase polymerase amplification (LF-RPA) assays were developed to detect the specific DNA sequences of *C. neoformans* and *C. gattii* in clinical CSF specimens [103,104]. The LF-RPA assay could detect 0.64 pg of genomic DNA of *C. neoformans* per reaction within 10 min and be highly specific to Cryptococcus species. The assay sensitivity was 95.2%, and the specificity was 95.8% [103]. Meanwhile, a separate LF-RPA assay was developed to amplify the capsule-associated gene, CAP64, of *C. neoformans* or *C. gattii* to detect cryptococcosis in CSF [104]. Nonetheless, while the LF-RPA assays provide a more rapid approach for screening cryptococcal meningoencephalitis, they cannot distinguish *C. neoformans* and *C. gattii*, and may not be able to detect cases where CSF is clear of infection.

Other recent advances include identification systems such as MALDI-TOF MS. This is a rapid identification tool that can identify both *C. gattii* and *C. neoformans* [105]. It offers a simple method for the separation of the eight major molecular types and the detection of hybrid strains within this species complex in the clinical laboratory. Nonetheless, a limitation of this technology is that it requires a positive culture and can only identify new isolates if the spectral database contains peptide mass fingerprints of the type strains of specific genera, species, subspecies or strains [106].

Surface-enhanced Raman scattering (SERS) and spectral analysis offer an additional potential diagnostic tool for *C. neoformans* and *C. gattii* [94]. This novel technology uses positively charged silver nanoparticles (AgNPs) as a substrate to distinguish *C. neoformans* and *C. gattii* in clinical samples directly. SERS is rapid and nondestructive and has relatively low equipment cost. Briefly, it was shown that AgNPs-self-assembled on the fungal cell wall surface via electrostatic aggregation, leading to enhanced SERS signals that were better than the standard substarte negatively charged AgNPs. The SERS spectra could then be used as a sample database in the multivariate analysis via orthogonal partial least-squares discriminant analysis. The SERS detection method can accurately (was shown to be 100%) distinguish between *C. neoformans* and *C. gattii* using principal component analysis. SERS seems to be a breakthrough, though further evaluation will be required before it can be introduced into routine clinical practice.

## 7. Conclusions

In conclusion, it is valuable to distinguish between *C. gattii* and *C. neoformans* at the individual patient level given their differences in patient predisposition, disease phenotype, and treatment approaches. Treatment trials should include patients infected with both strains and, where possible, run subgroup analyses to identify any differences.

At the population/public health level, molecular characterization of the strains circulating in specific geographies can provide valuable information on virulence and impacts on at-risk groups. Although there is currently no evidence of differences in MICs impacting treatment outcome it is important that in vitro and in vivo studies of new antifungals include both *C. gattii* and *C. neoformans*. Ongoing basic science research is vital to identify potential new targets for both organisms.

Ensuring that diagnostic tools are effective against both strains is also crucial. Further research comparing IRIS manifestations and therapies between the two species would be valuable in reducing morbidity and mortality.

The vast majority of high-grade evidence on the management of cryptococcosis comes from HIV-associated cryptococcal meningitis trials, which are increasingly conducted in low- and middle-income countries with the highest disease burden and a predominance of *C. neoformans* as the causative pathogen. Consequently, our understanding of cryptococcosis in other settings remains poor, and we struggle to know how much can be directly translated. These gaps can only be filled by establishing large international collaborative networks designed to describe the protean manifestations of cryptococcosis in well-established and emerging niche host groups. These large networks would be fertile ground for investigating novel antifungal or adjunctive therapies, including proper assessment of pharmacokinetic-pharmacodynamic and drug-drug interactions in disparate groups.

## Figures and Tables

**Figure 1 jof-09-00041-f001:**
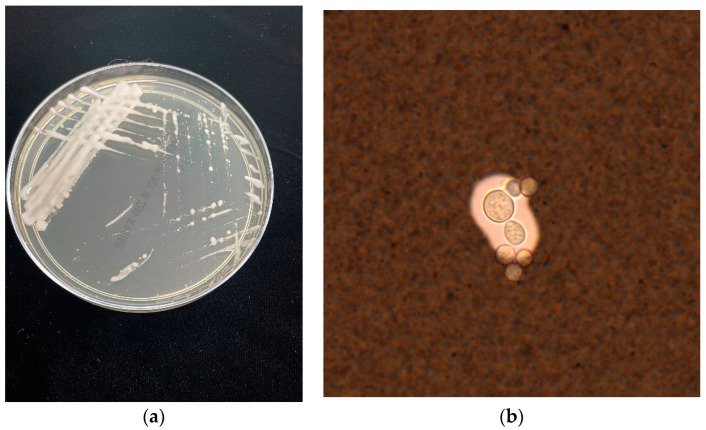
(**a**) Cryptococcus gattii growing on Sabouraud dextrose agar (**b**) *C. gattii* strained with India Ink.

**Figure 2 jof-09-00041-f002:**
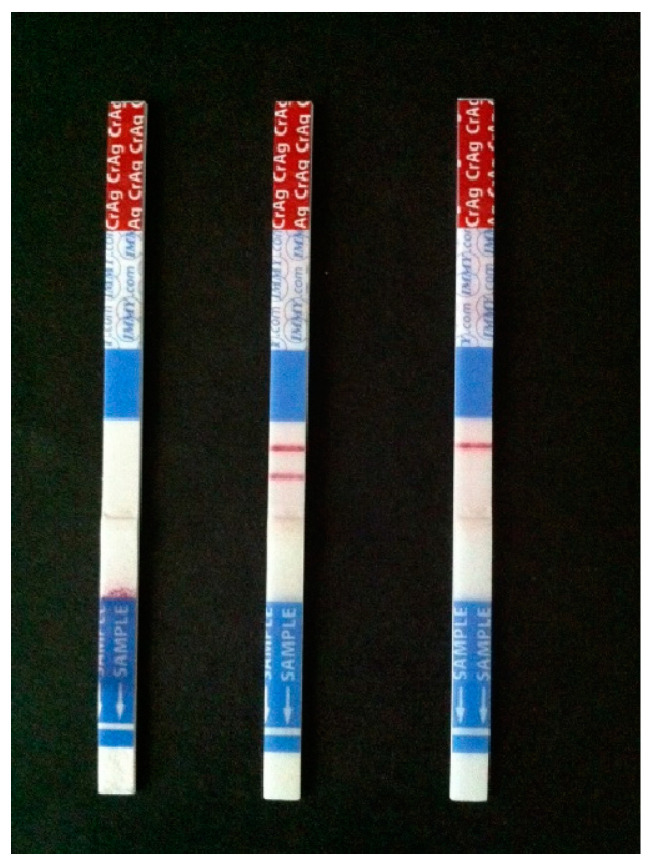
Cryptococcal antigen lateral flow assays (CrAg-LFA).

**Table 2 jof-09-00041-t002:** Comparison of morbidity, mortality, and antifungal susceptibilities between *C. gattii* and *C. neoformans* [4,5,6,7,39,50].

Pathogen	Hospitalization and Complications	Mortality by 12 Months	Antifungal Geometric Mean MIC
*C. neoformans*	Length of hospital stay: 2–210 days	PLWHIV: 20–61%	Fluconazole: 0.5–9.7 µg/mL
	Renal impairment: 28%		Voriconazole: 0.021–0.5 µg/mL
	Elevated ICP: 18%	Non-HIV: 8–50%	Posaconazole: 0.027–0.10 µg/mL
	Blindness: 12%		Amphotericin B: 0.098–0.69 µg/mL
*C. gattii*		10–43%	Fluconazole: 1.46–8.6 µg/mL
	Length of ICU stay: 1–29 days.		Voriconazole: 0.02–0.10 µg/mL
	Neurological sequelae: 17–27% at 12 months		Posaconazole: 0.04–0.36 µg/mL
	IRIS: 44%		Amphotericin B: 0.2726–0.39 µg/mL

Abbreviations: ICP, Intracranial pressure; ICU, Intensive Care Unit; IRIS, immune reconstitution inflammatory syndrome; MIC, mean inhibitory concentration; PLWHIV, person (s) living with Human Immunodeficiency Virus.

## Data Availability

Not applicable.

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
