# Peer review of "What’s New in Cryptococcus gattii: From Bench to Bedside and Beyond"

_jof, 2022, doi:10.3390/jof9010041_

Round 1

Reviewer 1 Report

  • A brief summary The authors gave narrative review summarizing the latest research on the taxonomy, epidemiology, pathogenesis, diagnostics and treatment of C. gattii infections.
  • General concept comments

    The authors reviewed the forementioned aspects of C. gattii infections adequately emphasizing the need for further international network research that will enable to include patients other than HIV patients and C. gattii as well (not just C. neoformans).

  • Specific comments 
  • line 130 - Several arboreal species have been identified as the predominant 130 environmental reservoir of C. gattii - it would be interesting to name several tree species from which C. gattii was isolated (for example, eucalyptus);
  • line 289 - authors should refer to acquired clinical resistance of cryptococcal strains to antifungal agents such as fluconazole; what is the current situation regarding this resistance in C. gattii and does it have impact on patient treatment and prophylaxis
  • in the Introduction section authors wrote: C. gattii is associated with 28 case fatality rates in range 10-25%. However, in the Table 2. range 10% to 43% for mortality by 12 months is mentioned. Please, explain.

Author Response

A brief summary The authors gave narrative review summarizing the latest research on the taxonomy, epidemiology, pathogenesis, diagnostics and treatment of C. gattii infections.

General concept comments

The authors reviewed the forementioned aspects of C. gattii infections adequately emphasizing the need for further international network research that will enable to include patients other than HIV patients and C. gattii as well (not just C. neoformans).

Many thanks for these comments

Specific comments 

line 130 - Several arboreal species have been identified as the predominant 130 environmental reservoir of C. gattii - it would be interesting to name several tree species from which C. gattii was isolated (for example, eucalyptus);

Thanks, we agree this is interesting. We have modified the text as follows, to provide some examples:

[addit] often eucalypts but including a huge range from oaks to baobabs

line 289 - authors should refer to acquired clinical resistance of cryptococcal strains to antifungal agents such as fluconazole; what is the current situation regarding this resistance in C. gattii and does it have impact on patient treatment and prophylaxis

Thanks for this comment. There was mention of the poor correlation between MICs and outcomes in the conclusion, but we have added a line to the body of the manuscript saying:

[addit at line 273] It is important to note that there is no evidence showing a relationship between susceptibility and treatment outcome.

in the Introduction section authors wrote: C. gattii is associated with 28 case fatality rates in range 10-25%. However, in the Table 2. range 10% to 43% for mortality by 12 months is mentioned. Please, explain.

The references giving the range 10-43% include late mortality (to 12 months) whilst 10-25% refers case fatality in the first 28 days, and is more conservative. We have adjusted the text and updated the refs for table 2 to clarify this, with thanks.

Reviewer 2 Report

This is very well organized overview of the Cryptococcus species, including C. gattii species complex and the clinical management of cryptococcosis. The information is timely, insightful and well balanced. I have only one suggestion for consideration. Although authors correctly highlighted several studies on the vaccine research, there are quite a few exciting additional reports on vaccine work hat authors may also want to mention. e.g., Stu Levitz lab has been developing glucan particle as a carrier and adjuvant for fungal vaccine, and several labs have reported Cryptococcus mutant strains as a potential vaccine candidates (sgl1, cda1/2/3, fbp1, H99gamma, et ac). Some of them have been found to provide protection against C. gattii challenge in rodent models. 

Author Response

This is very well organized overview of the Cryptococcus species, including C. gattii species complex and the clinical management of cryptococcosis. The information is timely, insightful and well balanced.

Many thanks for these comments

I have only one suggestion for consideration. Although authors correctly highlighted several studies on the vaccine research, there are quite a few exciting additional reports on vaccine work hat authors may also want to mention. e.g., Stu Levitz lab has been developing glucan particle as a carrier and adjuvant for fungal vaccine, and several labs have reported Cryptococcus mutant strains as a potential vaccine candidates (sgl1, cda1/2/3, fbp1, H99gamma, et ac). Some of them have been found to provide protection against C. gattii challenge in rodent models. 

Thanks also for this. The area of vaccine development in general, and adjuvants in particular, is a very interesting one. To do it proper justice is beyond the scope of our article, although we have added some specific comments about ongoing adjuvant work. We also took the chance to include a comments that clinical trials have occurred in other fungal pathogens and to provide readers with a reference to a review specifically on vaccine development for further reading.